# A Natural Zeolite Developed with 3-Aminopropyltriethoxysilane and Adsorption of Cu(II) from Aqueous Media

**Nasanjargal Shirendev [1], Munkhpurev Bat-Amgalan [1,2], Naoki Kano [3,*], Hee-Joon Kim [3], Burmaa Gunchin [4], Batdemberel Ganbat [5] and Ganchimeg Yunden [1,*]**

[1] Department of Chemical Engineering, School of Applied Sciences, Mongolian University of Science and Technology, Ulaanbaatar 14191, Mongolia

[2] Graduate School of Science and Technology, Niigata University, 8050 Ikarashi 2-Nocho, Nishi-ku, Niigata 950-2181, Japan

[3] Department of Chemistry and Chemical Engineering, Faculty of Engineering, Niigata University, 8050 Ikarashi 2-Nocho, Nishi-ku, Niigata 950-2181, Japan

[4] Institute of Chemistry and Chemical Technology, Mongolian Academy of Sciences, Ulaanbaatar 13330, Mongolia

[5] Department of Physics, School of Applied Sciences, Mongolian University of Science and Technology, Ulaanbaatar 14191, Mongolia

* Correspondence: kano@eng.niigata-u.ac.jp (N.K.); ganchimeg.yu@must.edu.mn (G.Y.); Tel.: +81-025-262-7218 (N.K.); +976-99074387 (G.Y.)

**Abstract:** In this work, we removed copper (II) from an aqueous solution by using zeolite modified with a silicon-organic monomer (3-aminopropyltriethoxysilane; APTES) depending on the pH, time, temperature, and initial concentration of Cu(II) ions. To confirm the modification process and assess the interaction between the modified zeolite and Cu(II), we performed instrumental analyses (XRD, SEM/EDX, TGA/DTA, BET, FT-IR, and XPS). We determined the maximum adsorption capacities of the modified zeolite for Cu(II) to be 4.50, 6.244, 6.96, and 20.66 mg/g at $T$ = 25 °C (pH = 5, $t$ = 8 h) when the initial concentrations of Cu(II) were 50, 100, 200, and 400 mg/L, respectively. According to the adsorption thermodynamics and kinetics, the second-order reaction controls the adsorption process. Based on the two isotherm models (Langmuir and Freundlich) with constant values ($K_L$ = 0.144, $n$ = 2.764) and the correlation coefficients ($R^2$ = 0.8946, $R^2$ = 0.9216), we concluded that the Cu(II) adsorption onto the modified zeolite could be followed by the Freundlich isotherm model rather than the Langmuir isotherm model. The modified zeolite could be an effective material for the removal of Cu(II) from aqueous solutions.

**Keywords:** adsorption; APTES (3-aminopropyltriethoxysilane); isotherms; heavy metal

## 1. Introduction

Nowadays, the production of water that contains heavy metals and chemicals is increasing due to rapid technological and industrial development. The primary producers of wastewater containing heavy metals are the leather, wool, cashmere, painting, mining, and metallurgy industries. Polluted water that contains heavy metals, including chromium, lead, copper, mercury, and nickel, can create ecosystem imbalances and cause cancer [1–3]. Heavy metal pollution in the environment usually follows the following cyclic order: from manufactory to the atmosphere, soil, water, and food to living organisms. When heavy metals enter the human body through food or water, they are acidified in the stomach. In this acidic medium, metals are oxidized to their various oxidative states, which can readily bind to biological molecules such as proteins and enzymes to form stable and intense bonds and free radicals. These can damage biological molecules. Therefore, the treatment of wastewater is necessary to protect the environment and human health [4,5]. Copper, one of these heavy metals, can cause brain and kidney damage in humans and living organisms.

Researchers have applied the chemical precipitation, ion exchange, precipitation, adsorption, membrane separation, and reverse osmosis methods for wastewater treatment. In recent years, studies on the separation of heavy metals using supported liquid membranes material have attracted many researchers' interest [6–8].

Of these methods, the adsorption method is the most selective and efficient if the cost of the adsorbent can be reduced [4]. Adsorbents are classified as follows, depending on the base material from which they are developed. These include adsorbents based on carbon, chitosan, and mineral (zeolite); magnetic adsorbents; biosorbents; and metal–organic or synergistic adsorbents [7,9–14].

The hydrated silicate mineral is one of the most efficient, low-cost, and environmentally friendly absorbent materials, and it can be used as a material for wastewater treatment [2]. Mesoporous aluminosilicates exist naturally and can be extracted synthetically. Zeolites are useful as catalysts for synthesis and in the polymerization of semiconducting materials and conducting polymers. The high surface area and pores of zeolites give zeolites unique adsorption properties. Zeolites have a large surface area due to their excellent porosity. The three-dimensional framework of $SiO_4$ and $AlO_4$ tetrahedra in the structure of zeolites is ordered and contains pores and spaces. Based on natural zeolite characteristics, composite materials with a high removal efficiency of organic, cationic, and anionic pollutants from wastewater have been created [10,15,16].

Mongolia has several deposits of natural zeolite reserves. The Tsagaantsav is one of the largest zeolite deposits in Mongolia, with 4.8 million tons of zeolite minerals, located 540 km from the capital city of Mongolia, Ulaanbaatar. Researchers have conducted several studies on the adsorption of heavy metals by using Mongolian zeolite due to its characteristics, such as its porosity, ion-exchange capacity, and reusability [17–19]. For example, researchers investigated the adsorptions of heavy metals from model aqueous wastewater by Tsagaantsav natural zeolite and have concluded that the cation-exchange capacity of the raw zeolite sample is proportionally related to the aluminum content [19]. Researchers studied the structural characterization and adsorption capacities of $Pb^{2+}$, $Cd^{2+}$, and $Zn^{2+}$ on the Tsagaantsav deposit and determined that the Tsagaantsav zeolites are highly effective at adsorbing $Pb^{2+}$ under acidic conditions, compared with $Cd^{2+}$ and $Zn^{2+}$ cations [20]. However, the practical application usage of natural zeolite is limited by its lower adsorption capacity and selectivity. In recent years, researchers have modified natural zeolites with functional-group compounds to increase the adsorption capacity and selectivity [21–28], with a few studies on improving the adsorption capacity of the Tsagaantsav natural zeolite.

On the other hand, Mongolia has abundant copper reserves, including the Erdenet, Oyu Tolgoi, and Tsagaan Suvarga deposits. Water pollution has attracted attention due to the rapid development of mining in Mongolia. Researchers have concluded that mining wastewater pollutes the groundwater around mining plants [29,30].

The adsorption properties of the zeolite-bead composite material and modified zeolite obtained from the Tsagaantsav deposit for Pb(II) and Cr(VI) were investigated [31,32]. However, studies are lacking on the Cu(II) adsorption properties of modified zeolites of Mongolia.

There are several reports in the literature on the adsorption properties of heavy metals onto silicates modified with organosilicon compounds [25,33–36]. However, there are not many reports regarding the rate-limiting step of the heavy metals onto modified zeolite. Determining the rate-limiting step of the adsorption process gives considerable information when considering the mathematical modeling equations used in the control of the process.

The objective of this study was to obtain the adsorption properties of Cu(II) on the zeolite from the Tsagaantsav deposit in the Dornogovi province of Mongolia, modified with 3-Aminopropyltriethoxysilane (APTES). We processed the pretreatment method based on the chemical and mineral compositions of the Tsagaantsav deposit. In addition, we focused on determining the rate-limiting steps of the adsorption process based on the kinetic parameters of the process.

## 2. Materials and Methods

### 2.1. Materials and Chemicals

We used zeolite from the Tsagaantsav deposit in the Dornogovi province of Mongolia. A raw sample with particle size 0.45–0.3 mm was used for the modification.

The chemicals and reagents used in the experiments were AR grade, and they were purchased from Damao Chemical (Tianjin, China).

A stock solution of Cu(II) (1000 mg/L) was prepared by dissolving $Cu(NO_3)_2$ into deionized water (>18.2 MΩ), treated by ultrapure water equipment (Barnstead Smart 2 Pure, Thermo Scientific, Atvidaberg, Sweden). The pH of the solutions of Cu(II) was adjusted by 0.1 M $HNO_3$ or 0.1 M $NH_4OH$.

### 2.2. Modification of Natural Zeolite

According to the elemental and XRD analysis, the zeolite from the Tsagaantsav deposit is thermally stable and dominated by clinoptilolite-K-type minerals. Based on this result, we heated the sieved zeolite sample for 3 h at 700 °C for the surface activation of the raw sample. Then, we washed the activated sample several times with hydrochloric acid solution (5%) and deionized water to eliminate impurities, and we then dried it for 2 h at 110 °C. After that, the activated zeolite was added to the modifier solution, 3-aminopropyltryethoxysilane (APTES) (volumes: 30%, 20%, and 10% in toluene), and we stirred the mixture for 3 h at a temperature of 50 °C. We filtered the suspension and washed off the solids with deionized water. After being dried at 110 °C for 4 h, modified zeolite was obtained. We cooled the modified zeolite in a vacuum desiccator until using it. We filtered the mixture and washed off the solids with deionized water. After being dried at 110 °C for 4 h, modified zeolite was obtained. We cooled the modified zeolite in a vacuum desiccator until used. We filtered the mixture and washed off the solids with deionized water. After being dried at 110 °C for 4 h, modified zeolite was obtained. We cooled the modified zeolite in a vacuum desiccator until used.

### 2.3. Adsorption Experiments

For each experiment, 0.1 g of adsorbent was dissolved in a 50 mL solution of a known amount of Cu(II) in a 250 mL Erlenmeyer flask, and we shook the suspension in a digital thermostatic stirrer. Then, we filtered the suspension, and the Cu(II) concentration in the filtrate was determined by ICP-OES (ICP-OES 7300 DV, PerkinElmer, PA, USA).

We conducted batch adsorption experiments at various pH levels, adsorption times, temperatures, and initial concentrations.

### 2.4. Characterization

To confirm the modification process and assess the interaction between the natural, modified zeolites and Cu(II), the phase identification of samples was performed using an X-ray diffractometer (D2 Phaser, Bruker, Billerica, MA, USA) in the 2θ range from 3° to 70°. Scanning electron microscopy and an energy-dispersive spectrometer (SEM6000-EDS2300, JEOL, Akishima, Tokyo, Japan) operating at 15 kV with various magnifications were used to observe the surface morphology and element distribution, respectively. Fourier-transform infrared spectroscopy (FTIR-4200, JASCO, Hachioji, Tokyo, Japan) was used to study the main functional groups that were present in natural and modified zeolites. The thermophysical and surface chemistry properties of samples were carried out by a thermogravimetric, differential thermal analysis (ThermoPlus2 TG8120, RIGAKU, Tokyo, Japan) and X-ray photoelectron spectroscopy (K-Alpha, Thermo Scientific Center, Waltham, MA, USA), respectively. The pore size measurements were estimated via the $N_2$-BET, TriStar II 3020 model (Micromeritics, Norcross, GA, USA).

### 2.5. Desorption Studies

For the desorption experiment, Cu(II)-adsorbed modified zeolite was transferred to an Erlenmeyer flask containing 50 mL of a desorbing agents, distilled water, and sulfuric acid,

respectively. The suspension was shaken at 100 rpm for 8 h. Then the modified zeolite was rinsed a couple of times with de-ionized water and added to Cu(II) solution for the next adsorption cycle. In each cycle experiment, the initial concentration of Cu(II) was 100 mg/L. Five-cycle adsorption/desorption test has been done. The percentage of desorption was found using the following equation.

$$\text{The percentage of desorption (\%)} = (C_{des}/C_{ads}) \times 100\% \tag{1}$$

where $C_{des}$ is the concentration of adsorbate desorbed (mg/L), and $C_{ads}$ is the concentration of adsorbate adsorbed (mg/L).

## 3. Results and Discussion

### 3.1. Characteristics of Natural and Modified Zeolite

A comparison of the instrumental analysis results (XRD, SEM/EDX, TGA/DTA, and $N_2$-BET) of natural and modified zeolite is shown below.

We present the mineralogical composition of the natural zeolite in Table 1. We could not identify some phases of the sample analysis by their interatomic distances in the X-ray diffraction analysis, and we have described them as unknown phases in Table 1. We determined the content of the unknown phases of the natural zeolite to be 7.9%. Natural zeolite minerals are complicated systems that contain many phases. The main minerals in the natural zeolite sample can be determined by X-ray diffraction analysis. The diffraction peaks with low intensity are observed by the side of the dominant diffraction peaks in the X-ray pattern. It is difficult to indicate the minerals corresponding to the diffraction peaks with low intensity in the original sample. Therefore, we noted the unknown minerals as x-phases.

**Table 1.** The mineralogical composition of natural zeolite.

| Mineral | Formula | Quantity (wt.%) |
|---|---|---|
| Clinoptilolite-K | $Ca_{1.16}Na_{1.8}Mg_{0.25}Al_{6.33}Si_{29.81}O_{72} \cdot 20.1H_2O$ | 60.3 |
| Quartz | $SiO_2$ | 2.7 |
| Goethite | $FeO(OH)$ | 4.7 |
| Anorthoclase | $(Na, K)[AlSi_3O_8]$ | 1.4 |
| Albite | $Na[AlSi_3O_8]$ | 15.2 |
| Hematite | $Fe_2O_3$ | 4.2 |
| Anorthite | $CaAl_2Si_2O_8$ | 2.9 |
| Unknown phases | - | 7.9 |

According to the XRD pattern in Figure 1, the diffraction peaks of the zeolite from the Tsagaantsav deposit appear in a range of 2θ angle values: 10.13 (100), 11.43 (200), 15.38 (220), 17.49 (111), 19.36 (131), 22.68 (400), 26.27 (222), 32.17 (530), 36.06 (351), and 37.24 (441). According to these diffraction peaks, the zeolite from the Tsagaantsav deposit is dominated by clinoptilolite-K-type zeolites [37–41].

According to these results, the crystal structure of the natural zeolite may have changed due to the APTES molecules interacting with the zeolite surface during the modification.

After the modification of the raw zeolite, the diffraction peak corresponding to clinoptilolite-K (d = 3.91 Å) shifted slightly (d = 3.90 Å) and the intensities of the diffraction peaks on the X-ray pattern were decreased.

We present the SEM images of the natural and modified zeolite in Figure 2a,b. As can be seen in Figure 1, the intensity of the diffraction peaks decreased after the modification of the natural zeolite. This means that the degree of crystallization of the crystal structure of the sample is decreasing. Consequently, these changes may be due to the irregular appearance of the surface of the modified zeolite compared to the natural zeolite, Figure 2.

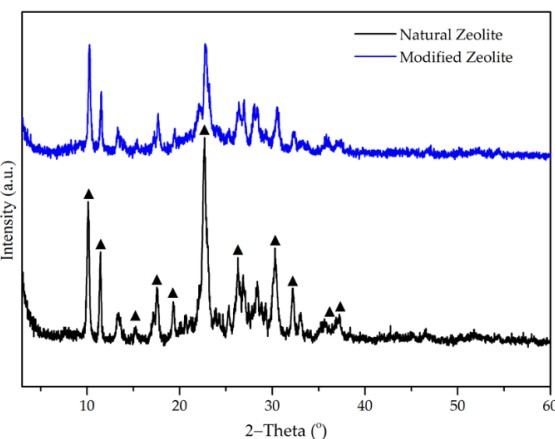

**Figure 1.** XRD patterns of natural and modified zeolite.

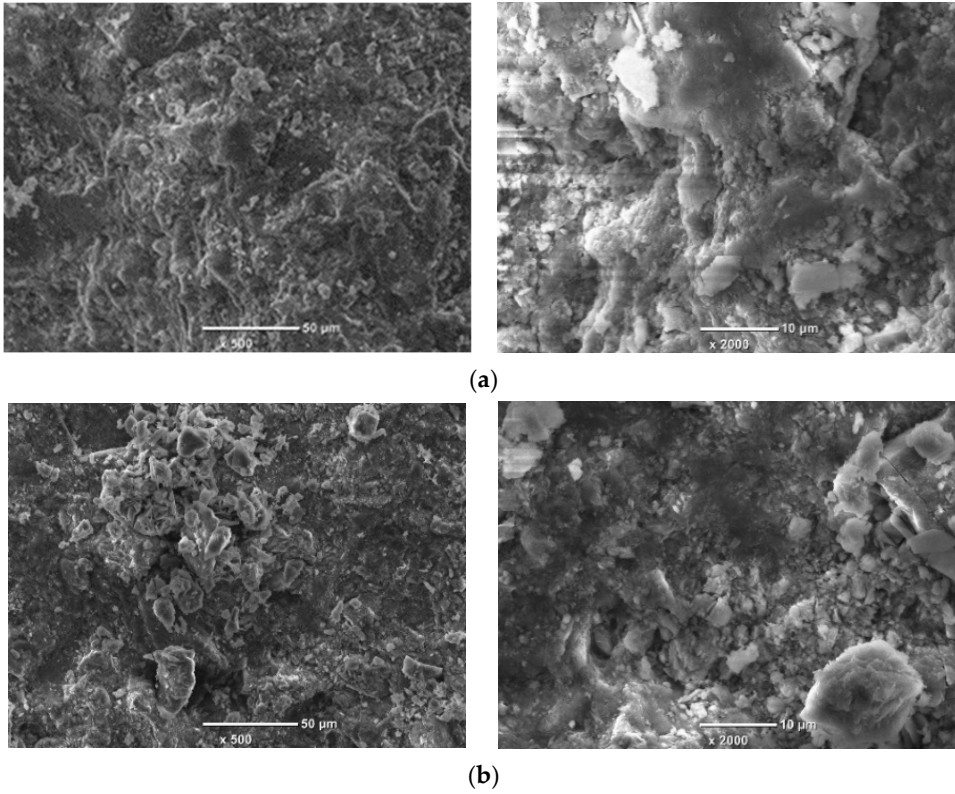

**Figure 2.** SEM images of (**a**) natural and (**b**) modified zeolite.

We present the results of the elemental analysis in Table 2. According to the ratio of silicon and aluminum (Si/Al = 5.9 mol/mol) and chemical composition (9.1% of potassium), the zeolite from the Tsagaantsav deposit is dominated by clinoptilolite-K-type minerals.

**Table 2.** Elemental analysis results.

| Elements | Quantity (wt.%) | |
|---|---|---|
| | **Natural Zeolite** | **Modified Zeolite** |
| Al | 10.8 | 11.0 |
| Si | 66.2 | 73.8 |
| K | 9.1 | 6.4 |
| Ca | 3.3 | - |
| Fe | 10.6 | 8.8 |

After the modification, the increase in the Si/Al ratio (by 0.6) and silicon content (by 7.3%) indicated that we could graft the groups of APTES onto the natural zeolite.

We conducted a thermal analysis of the natural and modified zeolites at a heating rate of 20 °C/min and in a temperature range from 20 °C to 1000 °C in a nitrogen gas dynamic atmosphere. We present the results in Figures 3 and 4. We present the TGA/DTA curves of the natural and modified zeolite to show the stages of composition in Figure 3 [42,43].

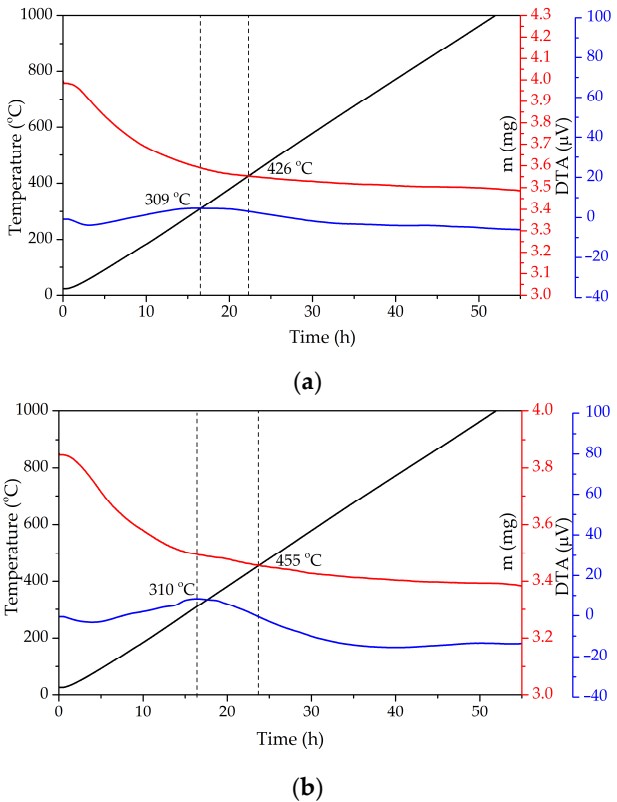

**(a)**

**(b)**

**Figure 3.** TGA (red curve)/DTA (blue curve) curves of (**a**) natural and (**b**) modified zeolite with temperature line (black line).

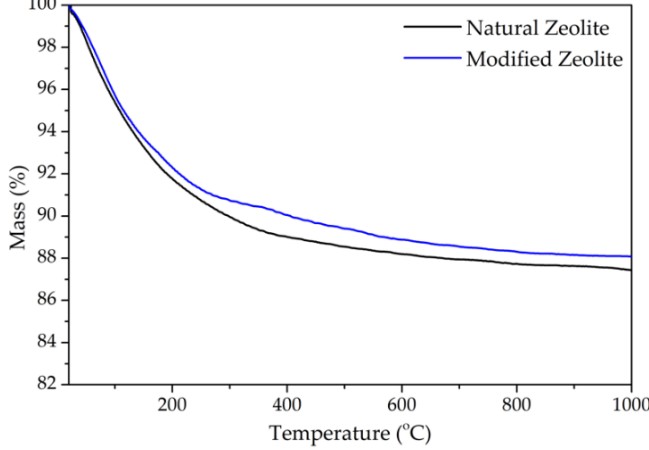

**Figure 4.** TG curves of natural and modified zeolite.

According to the TGA curve in Figure 3a, natural zeolite has two stages of composition. The first stage (from 20 °C to 426 °C) is for moisture and physically adsorbed water, with the water residing in the zeolite cavities and bound to the extra-framework-cation loss. The second stage (from 426 °C to 1000 °C) is for hydroxyl groups, and isolated OH groups

(structural water) were eliminated [42]. After the modification, the region of the first stage of combustion (from 20 °C to 426 °C) changed (from 20 °C to 455 °C), while we detected no significant difference in the exothermic peak (from 309 °C to 310 °C) in Figure 3b.

We compare the TGA curves of the natural and modified zeolites in Figure 4. The percentages of the masses of the solid particles after the complete combustion of the natural and modified zeolites were 87.43% and 88.1%, respectively (mass losses of 12.57% and 11.9%, respectively).

We can relate the difference in the mass loss (0.67%) to the silicon content of the APTES used for the modification. Based on the result of TGA/DTA, it can be concluded that, due to the presence of APTES in its composition, the modified zeolite is characterized by higher thermal stability than the natural zeolite.

The BET surface areas and the pore volumes of the natural and modified zeolite are shown in Table 3. The average pore diameter of the natural zeolite increased after the modification by 5.055 nm. Fuangfa Unob et al. [44] obtained similar results when the iron oxide was coated onto the silica support: the surface area decreased, and the average pore size of the resulting adsorbent increased. The increase in the average pore size may be due to the groups of APTES introduced onto the surface of the zeolite. After the modification, the degree of crystallinity decreases the specific surface area, and the pore volume of the modified zeolite decreases, but its average porosity increases. In addition, APTES molecules fill the internal pores of the natural zeolite surface, and its specific surface area and pore volume decrease, which leads to an increase in internal porosity (pore diameter). APTES possibly blocks the internal pores with small diameters of the zeolite and thereby modifies the surface of the silica. Consequently, we expected that the pore sizes in the modified zeolite would be larger, on average, than those of the natural zeolite.

**Table 3.** The surface properties of natural and modified zeolite.

| Sample | BET Surface Area ($m^2/g$) | Pore Volume ($cm^3/g$) | Average Pore Size (nm) |
|---|---|---|---|
| Natural zeolite | 22.79 | 0.058 | 10.188 |
| Modified zeolite | 9.06 | 0.035 | 15.243 |

We present the reaction of a silanol group on the zeolite surface with APTES resulting in the modified zeolite in Scheme 1.

**Scheme 1.** The functionalization of zeolite surface with APTES.

This functionality can change the surface properties and can be effective at improving the adsorption capacity, as shown below.

### 3.2. Adsorption Properties

#### 3.2.1. Effect of pH

We used APTES solutions of different concentrations (volumes: 30%, 20%, and 10% in toluene) for this modification. The adsorption efficiency of the zeolite modified by the APTES solution with a concentration of 30% was the greatest. Therefore, we used the zeolite modified by the APTES solution with a concentration of 30% as the adsorbent for all the experiments. We present the results of the effects of the solution pH on the adsorption of Cu(II) onto the natural and modified zeolite in Figure 5.

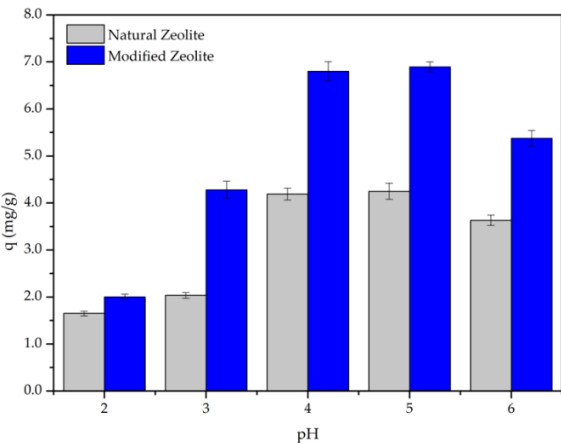

**Figure 5.** Effect of pH on the adsorption capacity ($C_0$ = 100 mg/L, $T$ = 25 °C, $t$ = 8 h, $n$ = 100 rpm).

The maximum value of the adsorption capacity of the modified zeolite was 6.06 mg/g at a pH of 5, which was higher than the adsorption capacity of the natural zeolite (4.05 mg/g), which confirms that we successfully performed the modification. When we increased the pH from 5, the adsorption capacity decreased the presence of the Cu(II) ion species in the aqueous solution. Therefore, we chose a pH of 5 as the optimal pH for the adsorption studies of the modified zeolite.

### 3.2.2. Effect of Initial Concentration and Time

We studied the effect of time on Cu (II) adsorption at a pH of 5 by using solutions with different initial concentrations of Cu(II). The result is shown in Figure 6.

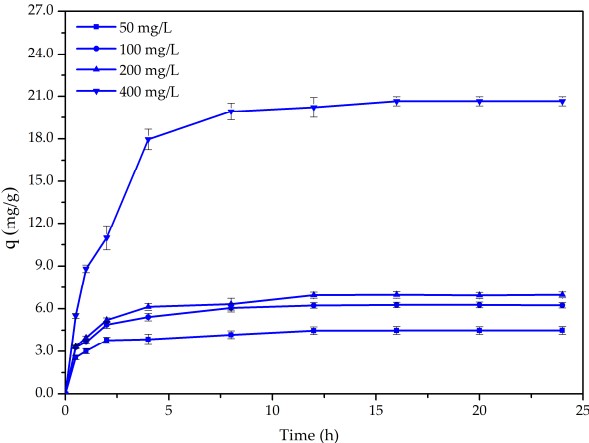

**Figure 6.** Effects of initial concentrations and time on the adsorption capacity (pH = 5, $T$ = 25 °C, $n$ = 100 rpm).

The increase in the adsorption capacity was rapid in the first 30 min, slowed during the next 8 h, and then became constant. The maximum adsorption capacity increased from 4.50 to 20.66 mg/g when the initial concentration of the Cu(II) increased from 50 to 400 mg/L. A higher value of the initial concentration gradient provided a significant driving force in the adsorption process and thus increased the adsorption capacity.

### 3.2.3. Effect of Temperature

We investigated the effect of temperature on the adsorption capacity of the modified zeolite at different temperatures, at an optimum pH, with different initial metal-ion concentrations (Figure 7). The change in temperature during the absorption process may have been affected by the interaction between the adsorbent and Cu(II), the stability of the

complex formed by the Cu(II), and the functional group of the modified zeolite and pore walls of the modified zeolite.

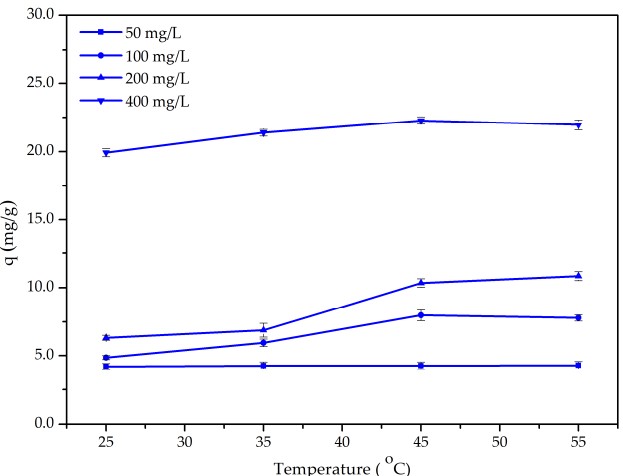

**Figure 7.** Effect of temperature on the adsorption capacity (pH = 5, *t* = 8 h, *n* = 100 rpm).

As shown in Figure 7, the adsorption capacity was increased with an increase in temperature from 25 °C to 45 °C, which is further evidence of an endothermic reaction. When the temperature increased from 45 °C to 55 °C, the adsorption capacity decreased slightly in some cases.

### 3.2.4. Effect of Competitive Cations

To study the effect of competitive cations, we conducted the experiment under an optimized pH (pH = 5) and adsorption time (i.e., 8 h), using different concentrations of common cations, as well as a combination of all four cations. Figure 8 shows that: the adsorption capacity decreased by approximately 0.5 mg/g in the presence of the competitive-cation-concentration increase.

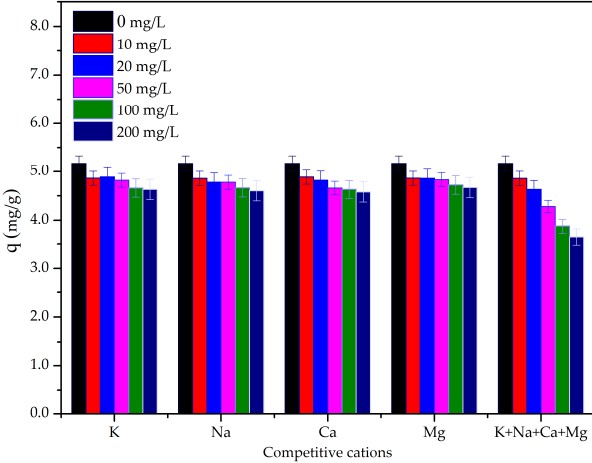

**Figure 8.** Effects of common cations on the adsorption capacity (pH = 5, *t* = 8 h, *T* = 25 °C, $C_0$ = 100 mg/L, *n* = 100 rpm).

We observed no significant decrease in the adsorption capacity when the concentrations of competitive cations were lower than 100 mg/L. The adsorption capacity was decreased by 1.3 mg/g (20%) for the combination of all four cations with concentrations of 100 and 200 mg/L.

### 3.3. Adsorption Kinetics

The effect of time on the adsorption can be expressed by the kinetic parameters. We applied the first- and second-order reaction and intraparticle diffusion models (Equations (2)–(4), respectively) to determine the kinetic parameters:

$$log(q_e - q_t) = logq_e - (K_1t/2.303) \tag{2}$$

$$t/q_t = (1/q_e)t + 1/(K_2q_e^2) \tag{3}$$

$$q_t = K_{id} \, t^{0.5} \tag{4}$$

where $q_e$ (mg/g) and $q_t$ (mg/g) are the denoted adsorption capacities at equilibrium and time ($t$); $t$ (min) is the time; $K_1$ (min$^{-1}$) and $K_2$ (g/mg·min) are the rate constants of first- and second-order adsorption; $K_{id}$ (mg/g·min$^{0.5}$) is the rate constant of the intraparticle diffusion.

The slope and the intercept of the linear gradient of the plot for the models were used to calculate the rate constants ($K_1$, $K_2$, and $K_{id}$) and calculated adsorption capacity ($q_e$, (cal)). We present the kinetic parameters in Table 4.

**Table 4.** The kinetic parameters of the adsorption.

| $C_0$ (mg/L) | $q_e$ (exp) (mg/g) | Pseudo First Order | | | Pseudo Second Order | | | Intraparticle Diffusion | |
|---|---|---|---|---|---|---|---|---|---|
| | | $K_1 \cdot 10^{-2}$ (min$^{-1}$) | $q_e$ (cal) (mg/g) | $R^2$ | $K_2$ (g/mg·min) | $q_e$ (cal) (mg/g) | $R^2$ | $K_{id}$ (mg/g·min$^{0.5}$) | $R^2$ |
| 50 | 4.501 | 2.213 | 1.459 | 0.9244 | 0.405 | 4.610 | 0.9997 | 0.411 | 0.8048 |
| 100 | 6.364 | 0.876 | 0.870 | 0.8122 | 0.257 | 6.452 | 0.9997 | 0.427 | 0.7805 |
| 200 | 6.945 | 2.412 | 2.454 | 0.8595 | 0.191 | 7.199 | 0.9994 | 0.749 | 0.7286 |
| 400 | 20.656 | 4.269 | 19.51 | 0.9246 | 0.033 | 22.173 | 0.9981 | 3.450 | 0.7934 |

For all concentrations of Cu(II), the regression-coefficient ($R^2$) values of the second-order model exceed 0.99. This result indicates the second-order model could describe the adsorption process well. Moreover, the amount of the calculated adsorption capacity ($q_e$, (cal)) from the second-order-model plot is relatively close to the adsorption capacity, determined by experiments ($q_e$ (exp)). Therefore, the second-order chemical reaction could be controlled by the process of the adsorption system.

### 3.4. Adsorption Thermodynamics

We evaluated the standard thermodynamic parameters (the standard Gibbs-free-energy change ($\Delta G°$), standard enthalpy change ($\Delta H°$), and entropy change ($\Delta S°$)) for the adsorption of Cu(II) on the modified zeolite at a temperature range from 25 to 55 °C.

$\Delta G°$, $\Delta H°$, and $\Delta S°$ are derived from Equations (5) and (6):

$$\Delta G° = -RTlnK_c \tag{5}$$

$$lnK_c = (\Delta H°/-RT) + (\Delta S°/R) \tag{6}$$

where $K_c$ is the equilibrium constant ($K_c = C_{Ae}/C_e$); $C_{Ae}$ is the concentration of adsorbed Cu(II) (mg/L); $C_e$ is the equilibrium concentration of Cu(II) (mg/L); $T$ is the temperature (K); $R$ is the universal gas constant (8.314 J/mol·K).

We found the $\Delta H$ and $\Delta S°$ from the slope and intercept of the linear plot ($lnK_c$ versus $1/T$). We summarize the results in Table 5.

We determined the values of the $\Delta G°$ in the temperature range of 298 to 328 K to be between −16.976 and −22.259 kJ/mol. According to the negative values of the $\Delta G°$, the adsorption of Cu(II) onto the modified is zeolite-favored and spontaneous. The increase in the negative values of the $\Delta G°$ with an increase in the adsorption temperature indicates that the process was more spontaneous at higher temperatures.

**Table 5.** The thermodynamic parameters of the adsorption.

| *T* (K) | *C*₀ (mg/L) | Δ*G*° (kJ/mol) | Δ*H*° (kJ/mol) | Δ*S*° (J/mol·K) |
|---|---|---|---|---|
| 298 | | −16.976 | | |
| 308 | | −18.674 | | |
| 318 | 100 | −21.894 | 40.914 | 194.498 |
| 328 | | −22.259 | | |

The standard enthalpy change Δ*H*° (40.914 kJ/mol) value is positive, a characteristic of an endothermic process. Moreover, the positive value of the standard entropy change Δ*S*° indicates the increased randomness at the solid–liquid interface.

*3.5. Adsorption Isotherm*

We obtained the adsorption isotherm for the Cu(II) on the modified zeolite at various weights of the modified zeolite (from 0.05 to 1 g) while maintaining all the other parameters (i.e., agitation speed (*n* = 100 rpm), pH (pH = 5), contact time (*t* = 8 h), initial concentration of Cu(II) (*C*₀ = 100 mg/L), and temperature (*T* = 25 °C)).

We employed the Langmuir and Freundlich isotherm models to describe the equilibrium metal adsorption.

The linearized form of these models is expressed by Equations (7) and (9), respectively.

$$C_e/q_e = (1/Qb) + (C_e/Q) \tag{7}$$

where $Q$ (mg/g) is the monolayer adsorption maximum capacity of the adsorbent, $q_e$ (mg/g) is the adsorption capacity at equilibrium, $C_e$ (mg/L) is the equilibrium concentration of Cu(II), and $b$ (L/mg) is the adsorption constant related to the energy of adsorption. $Q$ and $b$ were found from the slope and intercept of the plot of $1/q_e$ versus $1/C_e$.

The Langmuir model's dimensionless constant separation factor ($K_L$) is defined as:

$$K_L = 1/(1 + bC_0) \tag{8}$$

where $K_L$ indicates the shape of the isotherms: unfavorable ($K_L > 1$), linear ($K_L = 1$), favorable ($0 < K_L < 1$), or irreversible ($K_L = 0$).

$$logq_e = logK_f + nlogC_e \tag{9}$$

where $K_f$ (L/g) and $n$ are empirical constants of the Freundlich isotherm.

We calculated the isotherm model constants from the slope and intercept of the Langmuir and Freundlich isotherm model plots.

The values of the isotherm constants of these models and correlation coefficients are shown in Table 6.

**Table 6.** Isotherm constants.

| Isotherm Model | Isotherm Constant | |
|---|---|---|
| Langmuir | $Q$ (mg/g) | 6.176 |
| | $b$ (L/mg) | 0.064 |
| | $K_L$ | 0.144 |
| | $R^2$ | 0.8946 |
| Freundlich | $K_f$ (L/g) | 1.372 |
| | $n$ | 2.764 |
| | $R^2$ | 0.9216 |

Freundlich-model adsorption-intensity (*n*) values between 1 and 10 indicate beneficial adsorption.

Based on the isotherm-model constant values ($K_L$ = 0.144, $n$ = 2.764) and the correlation coefficients ($R^2$ = 0.8946, $R^2$ = 0.9216), we concluded that Cu(II) adsorption onto the modified zeolite could follow the Freundlich isotherm model rather than the Langmuir isotherm model.

**Table 7.** Comparison of the adsorption capacities of modified zeolite with some other adsorbents for Cu(II) adsorption.

| Adsorbent | Adsorption Capacity (mg/g) | Reference |
|---|---|---|
| Zeolite | | |
| Clinoptilolite zeolite | 4.9–15.4 | [47] |
| Chifeng zeolite | 4.0 | [48] |
| Synthetic zeolite | 9.5 | [49] |
| Modified zeolite | | |
| Natural alumosilicate modified with *N,N'*-bis(3-triethoxysilylpropyl)thiocarbamide | 29.7 | [25] |
| Natural zeolite modified with 4-(3-triethoxysilylpropyl) thiosemicarbazide | 29.5 | [33] |
| Dopamine functionalized tannic-acid-templated mesoporous silica | 58.7 | [45] |
| Silica and APTES–silica-modified $NiFe_2O_4$ | 63.5 | [46] |
| Zeolite modified with APTES | 4.5–20.6 | This study |

As shown in Table 7, a comparative result shows that the Cu(II) adsorption capacities of modified synthesized nanocomposite adsorbents [45,46] are higher than modified natural zeolites and alumosilicate [25,33]. 3,4-Dihydroxyphenethylamine (dopamine) and silicon organic compounds such as 4-(3-triethoxysilylpropyl) thiosemicarbazide, *N,N'*-bis(3-triethoxysilylpropyl)thiocarbamide, and 3-aminopropyltriethoxysilane (APTES) are used as a modifier agent in these studies. Sergey N. Adamovich et al. [33] determined the surface characteristics of Russian zeolite modified with 4-(3-triethoxysilylpropyl) thiosemicarbazide, which is more similar to the surface characteristics of Mongolian zeolite modified with 3-aminopropyltriethoxysilane (APTES). In addition, they concluded that chelate complexes formed in the Cu(II) ad-sorption system.

As can be seen in Table 7, the adsorption capacity of modified Mongolian zeolite was not necessarily high compared with other adsorbents. However, it could be regarded as a potential adsorbent for the removal of Cu(II) from wastewater for practical use; the modification method of the adsorbent is relatively simple, and it can be available for the removal of Cr(VI) [32].

*3.6. Interaction between Modified Zeolite and Cu(II)*

To discuss the reaction mechanism of the Cu(II) adsorption onto the modified zeolite, we applied characterization methods (FT-IR and XPS analyses) by comparing the results before and after adsorption. We present the results in Figures 9–11.

We found the FT-IR spectra of the modified-zeolite peaks in Figure 9 at 3646 cm$^{-1}$ (Si-OH), 3463 cm$^{-1}$ (OH) [38], and 1648 cm$^{-1}$ (for water molecules associated with the cations) [50]. We detected peaks for Si–O and Al–O bonds at 1052 cm$^{-1}$ and peaks for Si–O–Si and Al–O–Al bonds at 459 and 796 cm$^{-1}$, respectively. Moreover, we found peaks with low intensity at 2928 and 2858 cm$^{-1}$ (asymmetric and symmetric stretches of the CH bonds) [51], which indicated the presence of the propyl chains of APTES. The peaks appearing at around 1500 and 1600 cm$^{-1}$ belong to the $NH_2$ scissor vibrations, which indicated the presence of the $NH_2$ functional group of APTES [52]. These peaks correspond to the $Si(CH_2)_3NH_2$, which we detected with low intensity due to the low-concentration solution of APTES that we used for obtaining the adsorbent. After the Cu(II) adsorption, we detected the following changes in the FT-IR spectra: The peaks at 2928 cm$^{-1}$, 2858 cm$^{-1}$,

and 1513 cm$^{-1}$ that we recorded before the adsorption disappeared. The intensities of the peaks around 3400–3600 cm$^{-1}$ and at 1648 cm$^{-1}$ decreased, while the peak at 1063 cm$^{-1}$ (Si–O–Si) shifted to 1073 cm$^{-1}$. The Cu(II) interacted with a functional group of C, N, and O atoms, and the Si–O–Si-bond spectrum shifted due to the formation of metallic organic compounds.

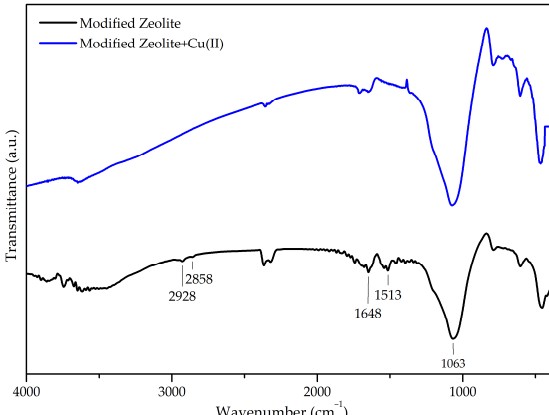

**Figure 9.** FT-IR spectra of modified zeolite, before and after the adsorption of Cu(II).

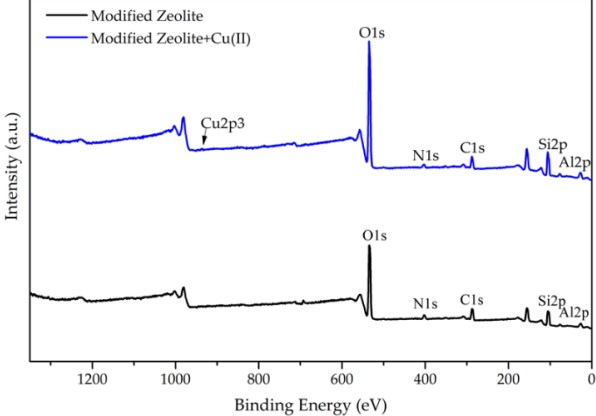

**Figure 10.** X-ray photoelectron spectroscopy (XPS) spectra of modified zeolite before and after the adsorption of Cu(II).

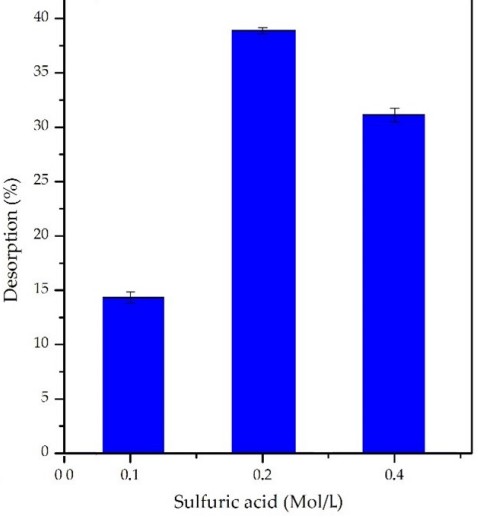

**Figure 11.** Desorption of Cu(II) from the modified zeolite.

We present the X-ray photoelectron spectroscopy (XPS) spectra of the modified zeolite before and after the adsorption of Cu(II) in Figure 10. As shown in Figure 10, scanned XPS spectra of the modified recorded the following peaks, 77 (Al2p), 105 (Si2p), 287 (C1s), 402 (N1s), and 534 eV (O1s), respectively [53–56].

According to the XPS results, as shown in Table 8, the main elements of the modified zeolite were silicon (26.55%), carbon (17.52%), oxygen (45.43%), and nitrogen (4.23%).

**Table 8.** Atomic ratio of the modified zeolite by XPS analysis.

| Atomic% | Before Cu(II) Adsorption | After Cu(II) Adsorption |
|---------|--------------------------|-------------------------|
| Al2p | 4.59 | 5.43 |
| Si2p | 26.55 | 27.5 |
| C1s | 17.52 | 12.42 |
| N1s | 4.23 | 2.38 |
| O1s | 45.43 | 52.2 |
| F1s | 1.68 | - |
| Cu2p3 | - | 0.07 |

The changes in the atomic values of O1s and N1s were more significant than other atomics after Cu(II) adsorption. It suggests that the Cu(II) interacted with the oxygen and nitrogen on the surface of the modified zeolite [57–60].

The XPS analysis was consistent with the results of the FT-IR, XRD, and EDX analyses.

Researchers [61,62] have proven the coordination interaction between heavy metals and the amine ($-NH_2$) and hydroxyl ($-OH$) groups. Based on all of the above mentioned results and previous studies on heavy-metal adsorption, the following reaction can be a controlled adsorption reaction in Scheme 2:

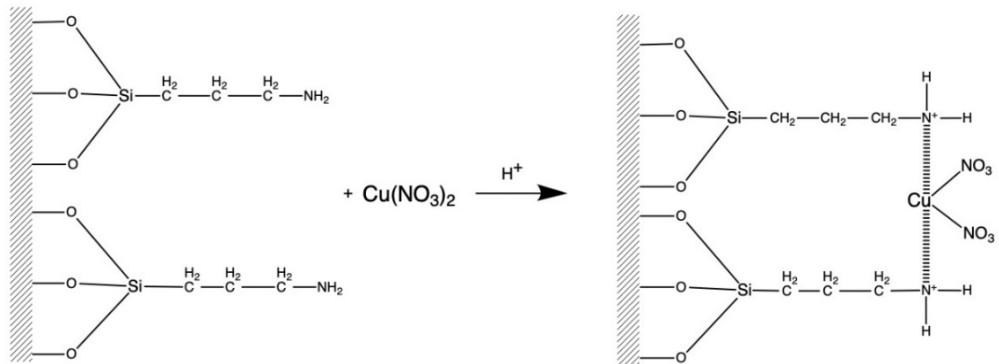

**Scheme 2.** Interaction between the surface of modified zeolite and $Cu^{2+}$.

### 3.7. Desorption studies

Stability is most important when the same adsorbent material is reused in multiple adsorption and desorption cycles. The experiment result for desorption studies using distilled water did not show any recovery of Cu(II). It indicates that the interaction between Cu(II) and modified zeolite is strong. Hence, sulfuric acid solutions (0.1, 0.2, and 0.4 mol/L) were used for the desorption study.

From the results of the desorption experiment Figure 11, it can be seen that the percentage of desorption is the highest in sulfuric acid solution with a concentration of 0.2 mol/L and that the maximum percentage of desorption was 38.9%.

These results indicated that Cu(II) is attached to functional groups of modified zeolite. In addition, 0.2 mol/L sulfuric acid solution was selected for the five-cycle adsorption/desorption test. From the experiment result of adsorption and desorption cycles in Figure 12, we can see that the desorption percentage was stable through 1–5 cycle processes. This result indicates that modified zeolite has good stability, although further investigation on regeneration study is needed.

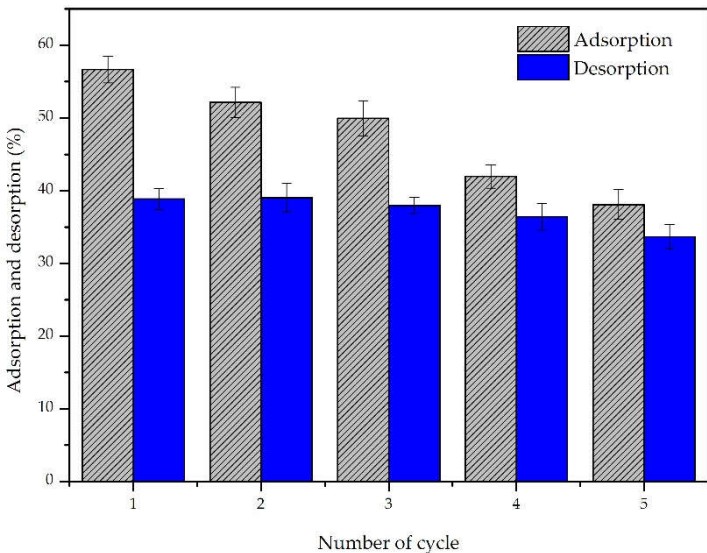

**Figure 12.** Regeneration of modified zeolite.

### 4. Conclusions

In this study, Mongolian natural zeolite was modified with APTES (3-aminopropyltrye-thoxysilane) successfully. The maximum adsorption capacities of the modified zeolite for the Cu(II) are 4.50, 6.244, 6.96, and 20.66 mg/g under optimal conditions (pH = 4.5, $t$ = 8 h, $T$ = 25 °C) when the initial concentrations of Cu(II) were 50, 100, 200, and 400 mg/L, respectively. The adsorption thermodynamics and kinetics confirmed that the rate-limiting step of Cu(II) adsorption onto the modified zeolite is the second-order reaction. The modified zeolite had a good adsorption efficiency for copper ions, which makes it a potential candidate in the field of wastewater treatment.

**Author Contributions:** Experiment and writing, N.S.; experiment and data evaluation, B.G. (Batdemberel Ganbat); instrument measurements, M.B.-A.; instrument measurements and data evaluation, B.G. (Burmaa Gunchin) and H.-J.K.; supervision and writing, G.Y. and N.K. All authors have read and agreed to the published version of the manuscript.

**Funding:** This research was supported by the "Higher Engineering Education Development" (M-JEED) Project (Research Profile code: J21C16). The present work was also partially supported by a Grant-in-Aid for Scientific Research from the Japan Society for the Promotion of Science (Research Program (C), No. 21K12290).

**Institutional Review Board Statement:** Not applicable.

**Informed Consent Statement:** Not applicable.

**Data Availability Statement:** Not applicable.

**Acknowledgments:** The authors thank Morohashi, H., and Amaki, Y., of the Industrial Research Institute of Niigata Prefecture for the XPS and TGA/DTA analyses and useful advice. The authors are also grateful to Teraguchi, M.; Nomoto, T.; Tanaka, T.; Hatamachi, T. of the Facility of Engineering; and to Iwafune, K. of the Institute of Research Promotion at Niigata University for the FT-IR, SEM/EDX, $N_2$-BET surface-area analyzer, and XRD analyses.

**Conflicts of Interest:** The authors declare no conflict of interest.

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
