# Peer review of "A Natural Zeolite Developed with 3-Aminopropyltriethoxysilane and Adsorption of Cu(II) from Aqueous Media"

_applsci, doi:10.3390/app122211344_

Round 1

Reviewer 1 Report (Previous Reviewer 2)

Reviewer’s Comments:

The manuscript “The Removal of Cu(II) from Aqueous Solution by Zeolite Modified with 3-aminopropyltriethoxylane (APTES)” is a very interesting work. In this work, we removed copper (II) from an aqueous solution by using zeolite modified with a silicon-organic monomer (3-aminopropyltriethoxylane) depending on the pH, time, temper- ature, and initial concentration of Cu(II) ions. To confirm the modification process and assess the interaction between the modified zeolite and Cu(II), we performed instrumental analyses (XRD, SEM/EDX, TGA/DTA, BET, FT-IR, and XPS). The results are consistent with the data and figures presented in the manuscript. While I believe this topic is of great interest to our readers, I think it needs major revision before it is ready for publication. So, I recommend this manuscript for publication with major revisions.

1. In this manuscript, the authors did not explain the importance of the Zeolite in the introduction part. The authors should explain the importance of Zeolite.

2) Title: The title of the manuscript is not impressive. It should be modified or rewritten it.

3) Correct the following statement “The Tsagaantsav deposit, which is located 540 km from Ulaanbaatar, is one of the largest zeo-lite deposits in Mongolia, potentially producing 4.8 million tons of zeolite minerals”.

4) Keywords: The keywords should be small. So, modify the keywords.

5) Introduction part is not impressive. The references cited are very old. So, Improve it with some latest literature like 10.3390/catal12101220, 10.3389/fmats.2022.1007485

6) The authors should explain the following statement with recent references, “We present the thermogravimetric analysis/differential thermal analysis curves of the natural and modified zeolite in Figure 3.

7) Please justfy the following statement “The second stage (from 426 to 1000 °C) is for the removal of the hydroxyl groups”.

8) The author should provide reason about this statement “The increase in the average pore size may be due to the groups of APTES introduced onto the surface of the zeolite”.

9) Comparison of the present results with other similar findings in the literature should be discussed in more detail. This is necessary in order to place this work together with other work in the field and to give more credibility to the present results.

10) Conclusion part is very long. Make it brief and improve by adding the results of your studies.

11) There are many grammatic mistakes. Improve the English grammar of the manuscript.

Author Response

Reviewer 2 Report (Previous Reviewer 1)

Accept as is.

In the experimental section, the authors should correct "N2-BET, TriStar II 3020..." to "N2-BET, TriStar II 3020..." .

Author Response

Reviewer 3 Report (New Reviewer)

The submitted manuscript is accurately presented and organized as a scientific text. The aim of this work is clearly presented and conclusions are based on the experimental results. The novelty of this work is based on the synthesis of an zeolite modified with a silicon-organic monomer but this study is focusing only in Cu sorption from synthetic waters. Nevertheless, as a preliminary steps for the full use of modified zeolite, this study deserves be published after some minor modifications.

1. Introduction part, if possible, some important and relative reports about adsorption could helped :https://doi.org/10.1080/01496395.2019.1676785,https://doi.org/10.1080/01496395.2016.1178289, https://doi.org/10.3390/app12157587

2. line 253, the authors must first determine the equilibrium time in order to fix it for the rest of the experiments

3. Why the choice of pH 05, it is preferable to determine the pH0 to be able to fix the ph of the work

4. I advise authors to spell English language throughout the manuscript. English is good, but some phrases are written with wrong word order. I don’t think it will take a lot of authors’ time.

5. The authors are requested to indicate how they calculate the b constant used in the calculation of KL.

6. In conclusions, what contribution could they make to understanding the behavior of the investigated modified zeolite concerning the adsorbed substrates?

 After these corrections, I consider the manuscript has the scientific level required to be published in the journal of applied sciences.

Round 2

Reviewer 3 Report (New Reviewer)

Accepted in present form

This manuscript is a resubmission of an earlier submission. The following is a list of the peer review reports and author responses from that submission.

Round 1

Reviewer 1 Report

Adsorption of metal ions has been investigated for decades using a variety of materials such polymers, silica, carbon materials, zeolites, clays, etc.

Particularly, silanization has been performed in this regard. In this sense, the paper provides no striking novelty, except that the authors employ a natural zeolite. In principle, this paper should be rejected, but I am here offering the opportunity to revise to address the following major points.

1) Improve the state of the art, and quote the very relevant, recently published paper:  J Taiwan Institute of Chemical Engineers, 2021, vol 129, 396-409.

2) Adsorption of copper on a silanized Mongolian zeolite does not suffice to claim novelty. This issue should be addressed in the Introduction.

3) Why consider only copper and not other heavy metal ions ?

4) The authors did not address the issue of competitive adsorption with heavy metals (for ex Zn2+, Pb2+, Cd2+...). This is very important to address. Does the adsorption of copper remains the same or does it decrease in the presence of another metal ion ? See for example the MDPI paper published by Lo et al. Sensors 202020(3), 580 . I understand competitive adsorption has been done in the presence of less toxic metal cations.

5) Table 7 is not interesting; it should compare silanized zeolites or silanized aluminosilicate materials. 

6) In general, adsorption is not that strong, and this should be discussed.

7)  The whole Cu2p doublet should be provided in Figure 11, and not Cu2p3/2. The peak shown is not centred at the right binding energy position. This must be fixed. Such a peak should show the main peak at 933-934 and shake-up satellites. See Surf. Interface Anal. 2017, 49, 1325–1334 for copper XP spectra, and Environmental Science and Pollution Research (2018) 25:20012–20022 for Cu2+ adsorbed by a sensor surface.

8) Silanization: XPS spectra of C1s and N1s before and after zeolite modification should be provided. 

9) The authors should discuss whether silanization alters the surface chemical composition of the zeolite (any change in Si/Al ratio ?)

10) Poor experimental details for characterization techniques in section 2.4. For example, XPS part indicates the K Alpha machine but not the source (Al Kalpha, 1486.6 eV), and no pass energy details. The authors have certainly used the charge compensation system but this is not mentioned.

To sum up , very well executed experiments but lack of novelty. The paper should be improved to stress what makes this piece of work unique, otherwise it will have no impact. 

Reviewer 2 Report

Reviewer’s Comments:

The manuscript “The Removal of Cu(II) from Aqueous Solution by Zeolite Modified with 3-aminopropyltriethoxylane (APTES)” is very interesting work. In this study, authors removed copper (II) from an aqueous solution by using zeolite modified with a silicon-organic monomer (3-aminopropyltriethoxylane) depending on the pH, time, temperature, and initial concentration of Cu(II) ions. To confirm the modification process and assess the interaction between the modified zeolite and Cu(II), we performed instrumental analyses (XRD, SEM/EDX, TGA/DTA, BET, FT-IR, and XPS). I believe this topic is of great interest to our reader, I think it needs minor revision before it is ready for publishing. So, I strongly recommend this manuscript for publication in this Journal with minor revisions.

1. In this manuscript, the authors did not explain the importance of Zeolite in the introduction part. The authors should explain the importance of Zeolite.

2. The author should provide reason about this statement “Polluted water that contains heavy metals, including chromium, lead, copper, mercury, and nickel, can create ecosystem imbalances and cause cancer”.

3. Introduction part is not impressive and systematic. Cite the following articles in the introduction part. (i) 10.3389/fchem.2022.975355 (ii) 10.1016/j.colsurfa.2021.127390

4. The author should provide reason about this statement…, We could not identify some phases of the sample analysis by their interatomic distances in the X- 120 ray diffraction analysis”.

5. The authors should explain regarding the recent literature why “he surface of the modified zeolite compared with the natural zeolite appears irregular, which may be due to the increases in the porosity that were the result of the introduction of APTES molecules onto the surface of the zeolite”.

6. In order check the stability of the synthesized materials, the authors should provide the SEM results after tests?

7. Comparison of the present results with other similar findings in the literature should be discussed in more detail. This is necessary in order to place this work together with other work in the field and to give more credibility to the present results.

8. The authors should provide stability of the synthesized materials.

9. The authors should pay more attention to the English grammar, and the abbreviation of journal names in Ref.

Round 2

Reviewer 1 Report

I agree on most revisions though I found the authors quite reluctant to revise.

The XPS section remains very weak. The experimental section has not been revised. Give source, whether it is monochromatic, indicate pass energy for the survey and narrow regions, indicate the step size, indicate if the charge compensation system has been used. Actually, the binding energies provided do not fit with the chemical states. And any chemical state given, requires display of the narrow spectrum. 

The statement "The peaks at 76.6 (Al2p), 104.59 (Si2p), 287.06 (C1s), 402.16 (N1s), and 533.88 eV (O1s) 393 could correspond to Al–OH, SiO2, Si–O–Si, C–N, N–O, and Si–O–Si, ..." should be removed, because the authors decline the suggestion to provide high resolution spectra and in the same time they provide chemical states that they CAN IN NO WAY deduce from the survey spectra. This is a non-sense and the journal IS NOT encouraged to publish these statements.  

So these values should be given for the peak position and NOT the chemical state. The values should be corrected because they are quite high. All of them are approximately 2-3 eV higher than should be. Peak positions taken from survey regions should be without decimals. 

The sentence should be revised as "77 (Al2p), 105 (Si2p), 287 (C1s), 402 (N1s), and 534 eV (O1s)." . All these values are a little bit high because the authors did not calibrate the spectra. They should calibrate using the C1s peak centred at 285 eV.

The statement "Compared with the spectrum before adsorption, the peak of Cu2p3 at 936.75 eV appeared after adsorption, which confirms that the adsorption was performed successfully" should be revised. The binding energy position is very high for Cu(II). 

This reviewer is willing to help the authors if the raw XY data are provided via the journal editorial office.

Reviewer 2 Report

The author responded to all issues, and I recommend accepting the manuscript in its present form.
